# The role of collegiality in academic review, promotion, and tenure

Diane (DeDe) Dawson[1]*, Esteban Morales[2], Erin C. McKiernan[3], Lesley A. Schimanski[4], Meredith T. Niles[5], Juan Pablo Alperin[6]*

1 University Library, University of Saskatchewan, Saskatoon, Saskatchewan, Canada, 2 Language & Literacy Education, University of British Columbia, Vancouver, British Columbia, Canada, 3 Facultad de Ciencias, Universidad Nacional Autónoma de México, México City, México, 4 Psychology Department, Capilano University, North Vancouver, British Columbia, Canada, 5 Department of Nutrition and Food Sciences, University of Vermont, Burlington, Vermont, United States of America, 6 School of Publishing, Simon Fraser University, Vancouver, British Columbia, Canada

* diane.dawson@usask.ca (DD); juan@alperin.ca (JPA)

**Data Availability Statement:** Survey responses can be found at the following publication: Niles, Meredith T.; Schimanski, Lesley A.; McKiernan, Erin C.; Alperin, Juan Pablo, 2020, "Survey

## Abstract

Review, promotion, and tenure (RPT) processes at universities typically assess candidates along three dimensions: research, teaching, and service. In recent years, some have argued for the inclusion of a controversial fourth criterion: collegiality. While collegiality plays a role in the morale and effectiveness of academic departments, it is amorphic and difficult to assess, and could be misused to stifle dissent or enforce homogeneity. Despite this, some institutions have opted to include this additional element in their RPT documents and processes, but it is unknown the extent of this practice and how it varies across institution type and disciplinary units. This study is based on two sets of data: survey data collected as part of a project that explored the publishing decisions of faculty and how these related to perceived importance in RPT processes, and 864 RPT documents collected from 129 universities from the United States and Canada. We analysed these RPT documents to determine the degree to which collegiality and related terms are mentioned, if they are defined, and if and how they may be assessed during the RPT process. Results show that when collegiality and related terms appear in these documents they are most often just briefly *mentioned*. It is less common for collegiality and related terms to be *defined* or *assessed* in RPT documents. Although the terms are mentioned across all types of institutions, there is a statistically significant difference in how prevalent they are at each. Collegiality is more commonly mentioned in the documents of doctoral research-focused universities (60%), than of master's universities and colleges (31%) or baccalaureate colleges (15%). Results from the accompanying survey of faculty also support this finding: individuals from R-Types were more likely to perceive collegiality to be a factor in their RPT processes. We conclude that collegiality likely plays an important role in RPT processes, whether it is explicitly acknowledged in policies and guidelines or not, and point to several strategies in how it might be best incorporated in the assessment of academic careers.

responses about review, tenure, and promotion", https://doi.org/10.7910/DVN/MRLHNO, Harvard Dataverse, V2, UNF:6:uj+VT99j/fw5qPomBc+ +qA== [fileUNF] Data regarding RPT documents can be found at the following data publication: Alperin, Juan Pablo; Muñoz Nieves, Carol; Schimanski, Lesley; McKiernan, Erin C.; Niles, Meredith T., 2018, "Terms and Concepts found in Tenure and Promotion Guidelines from the US and Canada", https://doi.org/10.7910/DVN/VY4TJE, Harvard Dataverse, V3, UNF:6: PQC7QoilolhDrokzDPxxyQ== [fileUNF].

**Funding:** Funding for this project was provided to JPA, MTN, ECM, and LAS from the Open Society Foundations (OR2018-46345). The funders had no role in study design, data collection and analysis, decision to publish, or preparation of the manuscript.

**Competing interests:** MTN is a member of the board of directors of The Public Library of Science (PLOS). This role has in no way influenced the outcome or development of this work or the peer-review process, nor does it alter our adherence to PLOS ONE policies on sharing data and materials.

# Introduction

Academic career progression in the United States and Canada is governed by review, promotion, and tenure (RPT) processes that typically assess candidates along three dimensions: research, teaching, and service. Although there is an increasing expectation that faculty should excel in all three dimensions [1], achievements in these three areas are not often weighed equally depending on the institution type, nor are they necessarily enough to guarantee a successful performance review or a promotion. Previous studies, including our own, have documented how research is often the most valued aspect of faculty work [2–6], with teaching second, and service activities a distant third [5, 7, 8]. However, it seems that even excelling in all three dimensions may not be enough. An additional, and controversial, characteristic—collegiality—has been the subject of robust debate [9–11], with some arguing for it to be added as a fourth dimension [e.g., 12,13], while others, notably the American Association of University Professors (AAUP), contend that if collegiality is to be assessed at all, it be within the three conventional categories [14].

There are two common understandings of the concept of collegiality. The first is captured by the Canadian Association of University Teachers (CAUT) that defines collegiality as the participation of academic staff in the collegial governance of the institution and states clearly that it "does not mean congeniality or civility" [15]. This form of collegiality is generally considered to reside under a faculty member's service obligations. A second understanding is aptly defined by Cipriano & Buller [16]: "Collegiality is instantiated in the relationships that emerge within departments and in the manner in which members of the department interact with and show respect for one another, work collaboratively in order to achieve common purposes, and assume equitable responsibilities for the good of the unit as a whole" (p. 46). Indeed, many in academia would acknowledge that a well-functioning department relies on the collaborativeness and constructive cooperation of its members. Supportiveness, respectfulness, and willingness to contribute all play a role in the morale and effectiveness of the academic department. In fact, research shows that these kinds of collegial behaviors contribute to institutional effectiveness [13, 17]. Collegiality among members of their department and/or the university was by far the most cited issue by faculty in a study of workplace satisfaction or dissatisfaction by Ambrose et al. [18].

While perhaps universally desired, collegiality is amorphous and subjective in nature, and thus difficult to assess fairly. In their statement, the AAUP notes that the inclusion of collegiality as a distinct criterion in RPT processes could be used as a cover for discrimination or to stifle dissent, effectively becoming a mechanism for enforcing homogeneity of thought or opinion to the detriment of the ideals of academic freedom for which tenure was established in the first place [14]. This is especially troubling if administrators attempt to intimidate or dissuade faculty from publicly questioning their decisions by accusing them of incivility or uncollegiality [19]. One response, from those who share these concerns but still support having collegiality assessed, could be to consider developing "equitable definitions of collegiality and clear measures that do not promote homogeneity, hinder academic freedom, or permit discrimination but that allow bad behaviour, such as bullying, to be addressed" [9] (p. 37). One step further could involve adopting instruments to assist in fair assessment such as the Faculty Disposition Rubric [20, 21], the Collegiality Assessment Matrix [10, 16], or a validated tool created by researchers at the University of Tampa to assess indicators of discretionary behavior [13, 22]. Whether or not collegiality is formally assessed, there is a common assumption that it plays an informal role in RPT decisions [see 17, 23].

Although not everyone agrees with a formal assessment of collegiality in the RPT process, collegiality matters in academic life. For one, individuals and instances that violate collegial

norms can disrupt the effective functioning of an academic unit, sometimes escalating to the point of bullying [see 24, 25]. Such instances have, at times, resulted in the dismissal of individual faculty members. In the U.S., faculty members who have been denied tenure based on a perceived lack of collegiality have often sued their universities for violating their rights, with limited success [for a description of various court cases see 9, 11, 26]. So far, the courts have consistently upheld the institutions' decisions in these cases, viewing these decisions as "an important factor in the ability of colleges and universities to fulfill their missions" [26] (p. 858) and have suggested that collegiality expectations be more formally included in employment contracts to provide clarity and avoid legal actions [13]. All this to say, when there are transgressions of collegial norms—perceived or real—there can be severe consequences for both individual academics and for the units and institutions they are a part of.

While some court rulings have advised institutions to incorporate collegiality expectations in their RPT documentation, it is unknown to what degree U.S. and Canadian universities have in fact adopted such policies, or whether they continue to follow the AAUP's recommendation to avoid explicit assessment of collegiality. Connell et al. [11] reviewed selected U.S. university policies that reference collegiality, and a very recent study by Lo et al. [27] explored the use of collegiality as a factor in librarian RPT documents at U.S. research intensive universities. However, we are not aware of any studies that have sought to analyze, across various institution types and disciplinary units, how current RPT guidelines at U.S. and Canadian institutions include the concept of collegiality. This study fills this gap by determining the extent to which the concept of collegiality (and related terms) is present in documents related to the RPT process. It is also unclear whether faculty perceive collegiality to be a factor in these processes, whether it is explicitly stated in their RPT documents or not. In doing so, we answer the following four related research questions:

1. Do faculty consider collegiality to be a factor in RPT processes?

2. How often do terms related to collegiality appear in RPT documents, and how do these vary across various institution types and disciplinary units?

3. How is the concept of collegiality defined within these documents?

4. To what extent and in which ways do RPT documents call for collegiality to be formally assessed?

## Methods

This study is based on the analysis of qualitative survey data collected as part of a project that explored the publishing decisions of faculty and how these related to perceived importance in RPT processes [see 28], and an analysis of 864 RPT documents collected from 129 universities from the United States and Canada and previously reported on in Alperin et al. [3] and McKiernan et al. [29]. Within this dataset of 864 documents are 381 unit-level documents from 60 of the 129 universities.

### Institution sample and document collection

The documents were collected from a representative sample of universities from the United States and Canada in 2016 and 2017. The sample of institutions was stratified based on institution type using the 2015 edition of the Carnegie Classification of Institutions of Higher Education [30] and the 2016 edition of the Maclean's University Rankings [31], which classify institutions into those focused on doctoral (i.e., research-intensive) programs (R-Type), those that predominantly focus on master's degrees (M-Type), and those focused on undergraduate

(i.e., baccalaureate) programs (B-Type). Following this strategy, we were able to obtain documents from 381 academic units of 60 universities (out of a set of 129 universities for which we obtained university-level documents). Full details of the sample selection and document collection strategy are available in Alperin et al. [3].

## Faculty survey

As described in Niles et al. [28], to develop the survey sample we searched for a page listing the faculty members at each of these 381 academic units (e.g., faculty, department, or school), and randomly selected up to five faculty members. We were able to identify 1,644 faculty from 334 of the 381 units spanning all 60 institutions (with some units not listing email addresses publicly, and some units not having five faculty members listed).

The selected participants were invited to participate in an online survey on September 17th, 2018, with reminders sent on a weekly basis until October 29th, 2018 to anyone who had not yet responded. A total of 338 people (22%) from 55 different institutions provided their written informed consent and proceeded to respond to the survey. Of these, 84 (25%) were faculty at Canadian institutions and the remaining 254 (75%) were from the United States; 223 (66%) were from doctoral research-intensive (R-Type) institutions, 111 (32%) from master's universities or colleges (M-Type) institutions, and 4 (1%) from baccalaureate colleges (B-Type) institutions. Responses were then anonymized, leaving only the institution type and discipline along with the survey responses for analysis, as per the research protocol filed with the Office of Research Ethics at Simon Fraser University (file number: 2018s0264).

In this paper, we report the results of two related questions from the survey that were previously unreported by Niles et al. [28]. The first question asked respondents to rank seven factors by their value in the RPT process. Of the 338 respondents to the survey, 268 respondents provided a full ranking of the factors presented to them. Ranked responses were counted using Microsoft Excel.

The second question was an open-ended follow-up asking respondents if there were any additional factors that they perceived as important for their RPT processes; 95 individuals provided responses which were then coded for the presence of the terms collegial or collegiality (*collegiality*), the presence of a similar term or concept (*collegiality-related*), and for other unrelated factors (*other*) (Table 1). Finally, some responses were coded as *non-answers* (i.e., respondents did not present a factor valued in the RPT process). All responses were coded by two independent coders (D.D. and E.M.) using the descriptions and examples in Table 1. A Kappa value of 0.89 was achieved for intercoder reliability [32]. The two coders met to discuss the seven answers (7%) where they differed and were able to come to an agreement on those codes.

**Table 1. Codes, definitions, and examples for the open-ended responses to this question in the survey: "Are there any other factors that you think are important for your review, promotion or tenure?".**

| Code | Definition | Example |
|---|---|---|
| Collegiality | The term collegial or collegiality is used. | "Perceptions of collegiality within a department play an "invisibly" large role." |
| Collegiality-related | Concepts related to collegiality (such as departmental citizenship, departmental politics, being likeable, having good relationships, professionalism) are expressed. | "Departmental citizenship. Rabble-rousers, complainers, and naysayers lose votes, I've seen it." |
| Other | Concepts not related to collegiality are presented. | "We are in a medical school, so—clinical practice excellence is needed." |
| Non-answers | Comments or other non-responses such as "no" and "not that I can think of". | "No—it is a pretty comprehensive list with the broad categories given." |

## RPT document querying, coding, and analysis

We loaded the full dataset of RPT documents into the NVivo 12 qualitative data analysis software [33] and queried the documents for collegiality and the related terms we identified through the analysis of the open-ended survey responses and from a review of the literature. We searched for the terms "collegial" or "collegiality" (resulting in 681 references across 228 documents), "citizen" or "citizenship" (resulting in 241 references across 134 documents), and "professionalism" (resulting in 103 references across 67 documents). While these are not the only terms that could be associated with the concept of collegiality, a preliminary reading of the RPT documents suggested they were the ones most commonly used to describe the concept, while other related terms (e.g., "respect") were primarily found in further descriptions or definitions of those three terms.

One person (D.D.) coded each of these references into one of three descriptive codes: *mentioned*, *defined*, and *assessed* using the descriptions and examples in Table 2. A randomly selected sample of 50 references was independently coded by another person (E.M.) to verify accuracy. A Kappa value of 0.89 was achieved for intercoder reliability. A fourth code was used

**Table 2. Codes, definitions, and examples for the qualitative analysis of the RPT documents.**

| Code | Brief Definition | Use this Code When: | Examples |
|------|------------------|---------------------|----------|
| Mentioned | Term is mentioned in context relevant to this study but without being defined and without details given on how it will be assessed. | • Instance is isolated use of the term (e.g., is mentioned in passing among a list of other desirable characteristics/behaviors in the candidate) **OR** | "Collegiality, cooperativeness, and willingness to mentor junior faculty would be important behavioral attributes." |
| | | • Instance consists of an example but no definition **OR** | |
| | | • Instance appears in a statement that it is considered or assessed in RPT processes but with no further elaboration (no definition or guidance on how to assess). | |
| Defined | More than a mention. The term is defined or elaborated upon, often with examples. | • Instance includes a definition or description of the term, and possibly a list of two or more examples of behaviors considered representative of the term **OR** | "Collegiality is more than civility and getting along with colleagues, staff, students and others in all university environments; rather it is consistent behaviors that show respect for others, cooperative and converted efforts to achieve department, college, and university goals, and the assumption of responsibilities for the good of the whole. Hallmarks of collegiality include, but are not limited to, cooperative interaction, open and honest communication, mutual support, respect, and trust of others, and collaborative efforts toward the common mission." |
| | | • Instance does not include a definition but includes enough specific examples of behaviors that the meaning of the term is clear. | |
| Assessed | More than a definition. Includes a description of how the term is going to be assessed in the RPT process. | • Instance includes instructions or suggestions to candidates on how to present evidence of the term (e.g., write a statement outlining your collegial behaviors) **OR** | "For each of the four areas of professional responsibility (teaching, scholarship, service, collegiality), tenured and tenure track faculty members will evaluate all other tenured and tenure track faculty members of the department, using the scale described below." |
| | | • Instance includes instruments or rubrics to assess the candidate on the term (e.g., a survey to distribute to colleagues, an assessment form or checklist, etc.) **OR** | |
| | | • Instance includes what will be considered as evidence of the term for RPT evaluation/assessment purposes. ***Note***: This goes beyond a list of example behaviours (such as in the "Defined" code), and includes clear direction that certain types of evidence will be used to assess the candidate on this term. | |
| | | ***Note***: If the document uses the word "assessment" but it doesn't describe how the term is assessed, then it is coded as either "Mentioned" or "Defined" | |

to identify mentions that used the key terms in ways and contexts that were not considered relevant for this study, which led to the exclusion of 338 (33%) irrelevant references [more details in the full codebook available in the accompanying dataset: 34].

Following the method described in Alperin et al. [3] and McKiernan et al. [29], we performed a "matrix coding query" to produce a table with institutions and academic units as rows, codes as columns, and a 1 or a 0 indicating whether the institution or academic unit made mention of the term or not, with the ability to distinguish if the mention appeared in documents that pertain to the whole institution, to one or more academic units, or both. We considered an institution as making mention of a term if the term was present in at least one document from that institution or any of its academic units.

## Results

### Research question 1: Do faculty consider collegiality to be a factor in RPT processes?

Overall, when asking respondents to rank the most important factor for RPT, they ranked research as the most important (mean 1.60), followed by teaching (mean 2.69), and grants (mean 3.33) (Fig 1).

The open-ended follow-up question in the survey asked whether there were other factors, not offered for ranking in the previous question, that respondents thought were important in their RPT processes. There were 95 respondents to this question, and all were from M-type or R-type institutions. Of these 95 responses, 39 (41%) indicated that collegiality or related concepts were a factor in RPT processes at their institutions (Table 3). This proportion was greater (55%) after discarding responses that did not contain a concrete proposal (e.g., those that said "none"). Of these 39, 12 respondents used the exact term *collegiality*, while a further 27

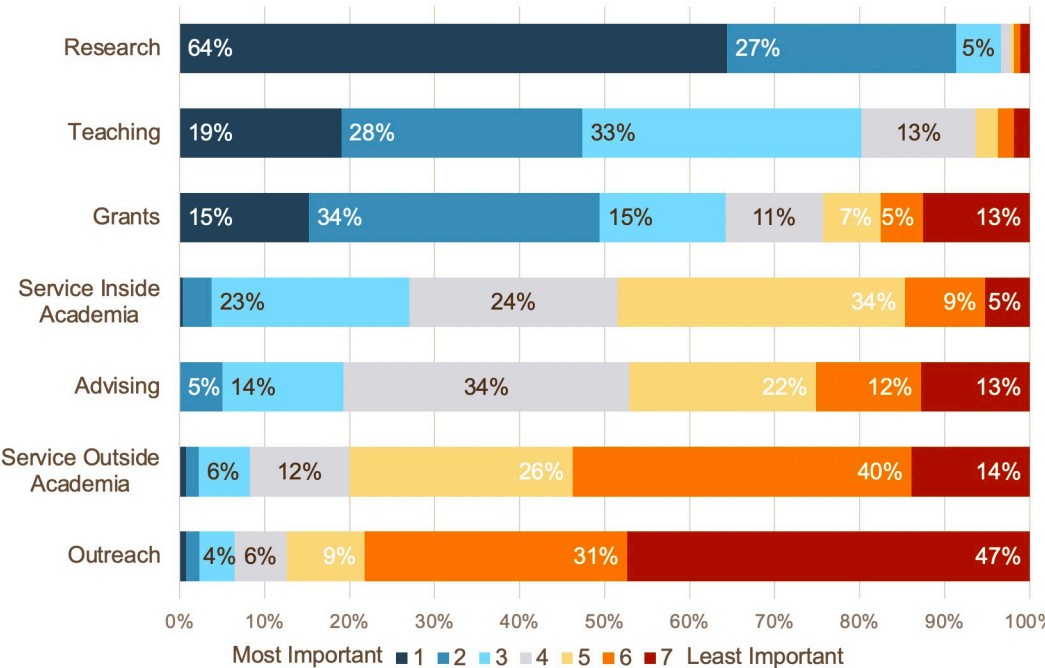

**Fig 1. Survey respondents' ranking of factors in response to the question: "Which of the following do you think is the most important for your review, promotion or tenure?".** Ranked in order of 1 (most important) to 7 (least important). Factors are ordered in their overall rate of importance (i.e., percent of respondents indicating a 1, 2, or 3).

**Table 3. Coding results for open-ended survey question: "Are there any other factors that you think are important for your review, promotion, or tenure?".**

| Code | M-type | R-type | Total |
|---|---|---|---|
| Collegiality | 4 | 8 | 12 |
| Collegiality-related terms | 7 | 20 | 27 |
| Other | 7 | 25 | 32 |
| Non-answers | 11 | 13 | 24 |
| Total | 29 | 66 | 95 |

N.B.: There were no respondents from B-type institutions to this question.

respondents used other language to describe *collegiality-related* concepts (see examples of each in Table 4).

## Research question 2: How often are terms related to collegiality referred to in RPT documents, and how do the references vary across various institution types and disciplinary units?

In the dataset of 864 RPT documents from 129 universities, the concept of collegiality (including related terms) was *mentioned* 507 times across 213 documents, *defined* 106 times across 85 documents, and *assessed* 51 times across 30 documents.

The concept of collegiality (including related terms) was *mentioned* across all types of institutions (R-Type, M-Type, and B-Type), and was *defined* or *assessed* only by a small proportion of R-Type and M-Type institutions (Table 5). These concepts were more prevalent at R-Type institutions (approximately twice as prevalent, when compared to M-Types), and least common at B-Type institutions, where they were mentioned only infrequently and never defined or assessed (Table 5). Within the R-Type institutions, the concept was *mentioned* most frequently (61%) in documents from Social Sciences and Humanities units but was *defined* and *assessed* more frequently in those from the Life Sciences (24% and 15% respectively) (Table 6).

To determine whether the frequency in which collegiality was mentioned across types of institutions and disciplines were significantly different from a uniform distribution, we used a chi-square analysis. The null hypothesis for both analyses was that the overall proportion of documents mentioning collegiality was the same between the different categories. The

**Table 4. A selection of open-ended survey responses coded as either *collegiality* or *collegiality-related*.**

| Collegiality | Collegiality-related |
|---|---|
| "Collegiality, integrity (academic and otherwise), general impression made on other faculty members and the tenure committees (departmental and faculty)." | "Playing the game. It's very much still personality based in many regards." |
| "We are also judged on "collegiality", which is nebulous due to a lack of clear policy on the criteria." | "To be likeable and not cause too many waves, especially if you are a person of color or women." |
| "Perceptions of collegiality within a department play an "invisibly" large role." | "The most important factor [is] internal politics and whether your colleagues like you. If they do, they will fight for your tenure regardless. If not, they will sink your tenure case." |
| "Yes, collegiality among Faculty and Students." | |
| "Collegiality." | "Departmental citizenship. Rabble-rousers, complainers, and naysayers lose votes, I've seen it." |
| | "Professionalism: internally with colleagues and participation in professional association." |

**Table 5. RPT documents' relevant references to collegiality and related terms by institution type.**

| | R-Type | | M-Type | | B-Type | |
|---|---|---|---|---|---|---|
| | N = 57 | | N = 39 | | N = 33 | |
| Mentioned | 34 | 60% | 12 | 31% | 5 | 15% |
| Defined | 17 | 30% | 6 | 15% | 0 | 0% |
| Assessed | 7 | 12% | 3 | 8% | 0 | 0% |

N.B. The conditions of the chi-square test were not met for the codes Defined or Assessed, but the chi-square analysis reveals the difference in the Mention of the concept of collegiality between institution types are significant. Chi-square tests: Code Mentioned: $\chi 2$ (2, N = 129) = 19.11, p<0.0001.

alternative hypothesis was that the proportion of documents mentioning collegiality was not equal across all different categories included in the test. Results show statistically significant differences (p<0.0001) in the number of times the concept of collegiality was *mentioned* across types of university (Table 5), but not when comparing across disciplines within the R-type institutions (Table 6) using a threshold of p>0.05.

Many of the RPT documents in the dataset that refer to collegiality or related terms do so either within the conventional three categories of research, teaching, and service (as advised by the AAUP), or else in an introductory section or preamble of the document. In most of these references, the term is mentioned in passing without being defined and without details given on whether or how it will be assessed. For example, the College of Education and Human Development at Texas A & M University-Corpus Christi has this sentence in the first paragraph of the preamble of their *Promotion and Tenure Policy*: "It is essential that faculty demonstrates dedication and achieves excellence in teaching, research/creative activity, **professionalism**, and professional contributions to preserve and strengthen the vitality of the university" [35] (p. 1). The term professionalism is not referred to again in the document.

In another example, this time from the University of Utah, Department of Political Science, the documents incorporate collegiality into a conventional Service category. The following statement appears in the introductory paragraph of the *Service* section of their *Policies and Procedures for Retention, Promotion, and Tenure of Regular Faculty* document: "Service is a fundamental part of being a member of the faculty of the Department of Political Science. The management and **collegiality** of the department depends on members of the faculty participating in the work of the department" [36] (p. 14). This is a typical example of the *mentioned* code in the dataset.

**Table 6. RPT documents' relevant references to collegiality and related terms in R-Type institutions by discipline.**

| | Social Sciences and Humanities | | Physical Sciences & Mathematics | | Life Sciences | | Multi-disciplinary | |
|---|---|---|---|---|---|---|---|---|
| | N = 38 | | N = 20 | | N = 33 | | N = 22 | |
| Mentioned | 23 | 61% | 8 | 40% | 14 | 42% | 8 | 36% |
| Defined | 6 | 16% | 4 | 20% | 8 | 24% | 4 | 18% |
| Assessed | 4 | 11% | 1 | 5% | 5 | 15% | 0 | 0% |

N.B. The conditions of the chi-square test were not met for the codes Defined or Assessed, but the chi-square analysis reveals the difference in the mention of the concept of collegiality between disciplines are not significant. Chi-square tests: Code Mentioned: $\chi 2$ (2, N = 113) = 4.46, p>0.05.

### Research question 3: How is the concept of collegiality defined within these documents?

We coded references to collegiality (and related terms) in the RPT documents as *defined* when the instances were more than a brief mention but do not go as far as discussing its assessment. In many of these instances the terms were defined or elaborated upon, often with or through examples.

The University of South Alabama, College of Arts and Sciences, provides this clear, concise definition with examples:

> Collegiality is more than civility and getting along with colleagues, staff, students and others in all university environments; rather it is consistent behaviors that show respect for others, cooperative and converted efforts to achieve department, college, and university goals, and the assumption of responsibilities for the good of the whole. Hallmarks of collegiality include, but are not limited to, cooperative interaction, open and honest communication, mutual support, respect, and trust of others, and collaborative efforts toward the common mission [37] (p. 4).

Whereas this example from Boise State University, Department of Psychological Science, begins with an explanation of the importance of collegiality with examples before giving a brief definition; although the definition is brief, the context provided with the examples makes the meaning clear:

> In addition, the Department values collegiality in the consideration of a candidate for promotion and tenure. Faculty members do not operate in isolation from other departmental colleagues. We must make decisions together regarding the undergraduate curriculum, class offering [,] student advising, the allocation of resources and space, and the hiring of new faculty members. These decisions require cooperation and professional interaction. None of these tasks can be successfully completed if each faculty member acts solely in his or her own personal interest. Collegiality emphasizes civility and reciprocal working relationships among professionals, as would be expected in any other workplace of a professional nature [38] (p. 1–2).

Institutions often use similar terms such as: respect, civility, and cooperativeness when defining collegiality. These terms focus on personality characteristics, whereas other institutions highlight desirable professional attributes. This is illustrated in the definition provided by Cameron University's Department of History and Government:

> Collegiality includes general professionalism in demeanor and appearance; a willingness to work with faculty, staff, and students on collective endeavors; a consistently demonstrated level of responsibility that includes prompt responses to email, telephone calls, and written correspondence, the submission of required administrative information, data, or reports on time, regular and prompt attendance at department or university meetings, knowledge of and adherence to all university policies, and a clear understanding of the proper professional line that should be drawn in faculty interactions with students [39] (p. 5).

Examples like this showcase the relationship between the terms *collegiality* and *professionalism*, the latter of which often appears within the category of teaching. Other related terms, like *citizen* or *citizenship*, usually occur within the service category, as in this definition from the University of Louisiana at Lafayette, Faculty Handbook which has a *Citizenship and Service* section:

The ideal faculty member is a model citizen of that community, helping to create an environment of collegiality. Such citizenship is manifested, for instance, in assuming administrative and leadership roles and in committee work at the department, college, and university levels. Institutional citizenship is displayed by assuming responsibility for improving the educational and research efforts of the institution, in counseling students about academic and personal matters, and in participating in the department's and University's outreach efforts in the community. Faculty are expected to treat all members of the campus community with respect and civility [40] (p. 6).

As in the examples above, when institutions include definitions of collegiality and related terms, they often do so within the existing categories of research, teaching, and service in adherence to the AAUP's recommendation. Several institutions refer directly to the AAUP statement [14], such as in this cautionary note from the University of Northern Colorado:

The requirement that review decisions (such as tenure and promotion) be based only on the results of comprehensive review in the areas of faculty endeavor (teaching, professional activity and service) precludes the use of collegiality as a separate dimension in making such decisions. The term collegiality has, historically, meant different things to different people. Sometimes, it indicates a legitimate concern for cooperativeness and team work. Sometimes, however, it has been used to foster an unhealthy uniformity of opinion that is a threat to academic freedom. The University of Northern Colorado adheres to the position of the AAUP by including the following note "On Collegiality As A Criterion for Faculty Evaluation" (November 1999). Collegiality should not be used as a separate category in reaching evaluative decisions. Where legitimate, it should be incorporated into the criteria for instruction, professional activity, and service [41] (p. 113).

## Research question 4: To what extent and in which ways do RPT documents call for collegiality to be formally assessed?

Some institutions went beyond mentioning and defining collegiality by providing some instructions or guidance on how it should be *assessed*. Formal assessment of collegiality in RPT documents is relatively rare, found only in the documents of only 8% of the institutions in our sample (12% of R-Type institutions, 8% of M-Type, and in none of the B-Type). The instances found in our sample ranged from suggestions to solicit statements from colleagues of the candidate to more formal likert scale evaluation forms distributed to colleagues.

For example, the *Tenure and Promotion Guidelines* at McNeese State University states that collegiality should be assessed through statements from colleagues. The guidelines read: "Statements concerning collegiality should be based on evidence of respect for peers, willingness to work toward departmental goals, professionalism and other such factors. Evaluations shall not be tainted by undocumented or hearsay evidence" [42] (p. 2).

At one institution, assessment of collegiality is considered by the *lack* of evidence to the contrary. The University of South Alabama considers a candidate's collegiality only during tenure processes and it is treated as a fourth criterion in these cases. The College of Education includes this statement in their *Tenure and Promotion Statement of Procedures and Criteria*: "The criteria are the same as for promotion plus the additional important consideration of collegiality with the Candidate's department. Absence of evidence and argument to the contrary will be considered evidence of the Candidate's collegiality with the department" [43] (p. 2). The College of Engineering includes very similar language [44]. Whereas the Pat Capps Covey

College of Allied Health Professions includes this question, under its own category specific for Collegiality, for reviewers of the tenure candidate's file to consider: "Is the applicant compatible with colleagues in the Department?" [45] (p. 45).

Other universities take a mixed approach, taking both evidence and lack of evidence into account. For example, the University of Southern Mississippi, College of Education and Psychology, Department of Child and Family Studies, states in its *Tenure and Promotion Guidelines* that "Candidates are expected to demonstrate a continuing pattern of respecting and working well with peers, students, staff, and the unit's common purpose. Collegiality will be evaluated by the presence of a variety of positive indicators and the absence of negative indicators. Candidates are encouraged to address the issue of collegiality in the narrative they provide for review" [46] (p. 7). The document goes on to provide a reasonably comprehensive, though not exhaustive, list of specific examples of positive and negative indicators of collegial behaviors. Several other departments within the College of Education and Psychology include similar language in their RPT guidelines. Interestingly, although these departments are quite thorough in defining and providing guidance on the assessment of collegiality, they stop short of explicitly listing it as a fourth criterion. Other Colleges and Departments at the University of Southern Mississippi similarly defined collegiality but were explicit in indicating that it should not be considered a distinct performance category.

Contrary to the recommendations of the AAUP, some institutions or units treat collegiality as a fourth criterion in their RPT processes. The Southern Utah University (SUU), for example, provides what are arguably the most thorough guidelines for assessing collegiality from the documents in our sample. The guidelines are based on a university-level policy [47] that outlines faculty professional responsibilities to students, colleagues, and the institution, as well as disciplinary actions if the faculty member fails to meet the responsibilities. However, various departments within SUU assess it in different ways. For example, the Department of Accounting assesses faculty in each of the four categories yearly in a Faculty Annual Activity Report (FAAR). For the fourth category of *Collegiality*, candidates must demonstrate "Full compliance with SUU Policy 6.28 (latest edition) and achieve a five-year average score on collegiality of 0.80 from an anonymous survey of all department faculty members. The survey is completed by all department faculty members at the start of each academic year, and uses a two-point scale (0 = not collegial, and 1 = collegial)" [48] (p. 2). The Biology Department, on the other hand, requires candidates to write a summary statement of their collegiality for their RPT dossiers and provides a faculty survey for colleagues to assess the candidate using a 5 point scale on specific collegial attributes under the headings of "relationships with others" and "institutional citizen" [49] (p. 8). And finally, the Psychology Department uses a Department Evaluation of Peers document wherein each faculty member assesses all other faculty members along a scale from Unacceptable to Meritorious in each of the four criteria of teaching, scholarship, service, and collegiality [50].

## Discussion

Our survey of faculty revealed that beyond the typical criteria related to research, teaching, and service commonly evaluated in RPT processes, there are clear signs of an additional focus on the more intangible characteristic of collegiality. Among respondents who provided additional factors important for the RPT process, collegiality was the most common additional factor, suggested more often than all other responses combined. In searching for this concept in the RPT documents in our sample, we found that the prevalence of the related terms varied widely across institution types, appearing in the documents of only 15% of B-Type institutions in our sample, but in 60% of those from the R-Type institutions. Notably, while collegiality was

mentioned in many of the RPT documents, far fewer defined the term and even fewer explained how collegiality was assessed. This apparent simultaneous reliance on, but ambiguity surrounding, the concept of collegiality could introduce potentially problematic subjective criteria and even bias into the RPT process by evaluators applying their own definitions of the concept [51].

Despite the overall prevalence of mentions, we found that it is rare for institutions to specify collegiality as a formal fourth criterion for evaluation in RPT documents. The majority of universities appear to adhere to the AAUP recommendation: if they refer to collegiality or related terms at all, it is usually within the three conventional categories (research, teaching, and service) or in a broad preamble statement. However, some authors have observed a growing trend in the use of collegiality in academic evaluations. In 2001, Connell and Savage reviewed the relevant U.S. court cases noting that "...courts have affirmed at every turn the use of collegiality as a factor in making decisions concerning faculty employment, promotion, tenure, and termination ..." concluding that universities "...should feel confident in considering collegiality in faculty decisions and that it is unnecessary for them to specify collegiality as a separate and distinct criterion" [26] (p. 858). In their follow-up study ten years later, Connell et al. [11] note that the trend of courts siding with institutions continues and that there is also an increase in universities "...using collegiality in making important employment decisions ..." and adopting statements or policies regarding this (p. 572). Little appears to have been written about this in the Canadian context; we suspect this may be because most disputes are handled by appeal or grievance within the university and do not make it into the court system.

When collegiality or related terms are referred to in the RPT documents in our dataset, they are usually just mentioned briefly or in passing without further explanation or definition. This resonates with the findings of Lo et al. [27] who looked at the prevalence of collegiality and related terms in RPT documents of librarian faculty at R-Type universities in the United States: of the approximately one-third of institutions in their sample that mention collegiality in their RPT documents only a small number of these actually define the term or specify how it should be assessed. Connell et al. [11] also found that when institutions make reference to collegiality they usually do so "...briefly or broadly in their tenure and promotion policies or faculty handbooks, but do not include it as a separate criterion for review" (p. 570). Briefly mentioning that collegiality is an important consideration, but not defining it, or outlining how it is to be assessed, potentially opens it up to being misinterpreted or abused in RPT decisions, potentially more than if RPT documents do not discuss collegiality at all. The danger is that the concept of collegiality can be weaponized to eliminate perceived "troublemakers" or those who do not "fit in" for various reasons. The concept of collegiality is highly subjective, but it can be argued that the terms and concepts used in assessing research are also subjective and lack clear definitions [52–54]. And Connell & Savage [26] agree: "Although collegiality is a vague and subjective term, there is no question that evaluation of scholarship, research, and teaching is also very subjective" (p. 854). Despite the fact that collegiality is poorly defined, or not defined at all in RPT documents, some faculty still perceive that it plays a role, as the responses to the open-ended question in our survey reported in this study indicate.

The majority of respondents who provided answers to our survey question about other factors considered in RPT decisions were from R-Type institutions, which is commensurate with our response distribution of institutional types. Of the 39 responses that indicated collegiality or related factors were considered, 28 (71%) were faculty from R-Type universities (as compared with 66% of the respondents), which corroborates our RPT analysis where collegiality and related terms were most prevalent in R-Type institutions.

That R-Type institutional respondents were most likely to mention collegiality is likely both a function of our distribution of respondents but also may be related to the nature of R-Type

institutions. R-Type institutions are research intensive, as compared with M-Type or B-Type institutions, which may affect collegiality perspectives and experiences. In *Generous Thinking: A Radical Approach to Saving the University* [55], Kathleen Fitzpatrick identifies an individual-istic and hyper-competitive environment of research universities as a key factor in faculty burnout and the undermining of collaborative relationships among colleagues. She further argues that RPT processes may be presented as meritocratic but "[i]n actual practice, however, those metrics are never neutral, and what we are measured against is far more often than not one another—sometimes literally" [55] (p. 26). Indeed, increases in academic workload and time demands in the areas of teaching and research, and the focus on performance metrics that recognize these activities, results in increased competition among colleagues and dimin-ished attention to academic citizenship behaviors [56]. Such an environment can fuel resent-ments and disrespectful conduct among colleagues. These "perverse incentives" and a "pervasive culture of competition" actively discourage faculty from engaging in activities that would facilitate or contribute to the success of their colleagues [57]. Another possibility for the mentions of collegiality in R-Type institutions, might be that these institutions have recognized the value of collegiality, which has led to the inclusion of the concept into evaluations. That is, that the greater presence of collegiality in the evaluation process and documents reflects the value placed on it by these institutions. Such an interpretation would be in opposition to Fitz-patrick, Agate, and their colleagues' view of these institutions, but would align with the reality that collaboration is implicitly incentivized on the research track through the citation advan-tage of multi-authored publications [58, 59]. Finally, it is also possible that the inclusion of col-legiality in documents is a function of the size of the institutions and of the academic units within them, something our study did not test. It may be that the relatively larger size of R-Type institutions requires collegiality to be managed institutionally, while collegial relation-ships emerge more organically in smaller groups.

## Conclusions

The results from our survey respondents in the United States and Canada suggest that the con-cept of collegiality plays a role in RPT decisions, even at institutions that do not explicitly acknowledge it as a factor in their processes or guidelines. This role may be indirect or infor-mal as is suggested by the lack of definitions and assessment in RPT documents demonstrated through our assessment. However, despite the potential informal nature of collegiality in the RPT process, it must be acknowledged if we are to take seriously the concerns about the unfair influence of departmental politics, biases, and personal grievances that have emerged through court cases [9]. Acknowledging this role does not necessarily mean elevating collegiality to its own distinct criterion in the RPT process, which the AAUP [14] warns poses several dangers such as promoting homogeneity of thought, discouraging dissent, and acting as a cover for dis-crimination. Instead, universities or units could incorporate some kind of systematic approach to address collegiality within their existing evaluation frameworks that typically include the tri-fecta of research, teaching, and service. This could mean developing "clear definitions of teach-ing, scholarship, and service, in which the virtues of collegiality are reflected" as advised by the AAUP [14]. Encouraging collegial behaviors in this manner has the potential to improve the morale and job satisfaction of faculty while also increasing the overall effectiveness of the unit [13, 18].

While collaborative and collegial behaviors are necessary for the effective functioning of an academic unit and the contentment of its faculty, we must also recognize that it is complex to fairly assess collegiality, either as a criterion in RPT processes or as a dimension of other activi-ties. Perhaps a values-enacted approach to assessment, as exemplified by the HuMetricsHSS

initiative (https://humetricshss.org/) and discussed by Agate et al. [57], may present a viable means to include the aspects of collegiality that are desirable within a larger evaluation framework. As per Agate et al.'s approach, "values-enacted indicators" could be developed by each institution, or unit therein, to align with the core values or mission of the group, and a subset of these indicators, which they refer to as "vicarious indicators", could be used to recognize faculty who facilitate the success of colleagues through activities such as mentorship or providing formative reviews. Such an approach could reward this kind of traditionally undervalued labor while encouraging collegiality and collaboration. Agate et al. [57] note that this kind of evaluation is not unknown in the academy; administrators are often assessed on the success of those they lead. While none of the institutions that defined or assessed collegiality used a value-centric approach, there is ample opportunity for them to do so, especially as momentum continues to build towards research assessment reform [60–63].

## Limitations

There are several limitations to the findings in this study, similar to the limitations mentioned in the previously published articles on this survey dataset [28] and this RPT document dataset [29]. Both the survey and the RPT documents have a geographic focus of Canada and the United States. We acknowledge that this means the findings are likely not representative of other regions globally. Additionally, the survey responses rely on the participants' self-reported information and perceptions of the importance of collegiality in RPT processes at their institutions. This may not align with the experiences of their colleagues or the stated practices of their units.

The types of documents collected in the RPT dataset are diverse: from university-level faculty handbooks to department-level standards and guidelines for RPT assessment and processes. As such, some of these documents contain more specific information than others regarding expectations of candidates. As such, the lack of presence of collegiality or related concepts may be due to the types of documents used at those institutions or assembled in our dataset, and not a lack of interest or focus on using this criterion for evaluation. Finally, studying the RPT process through a document-centric approach such as this limits our analysis to what is formalized in the documents themselves. This approach is further limited by the terms which we chose to include in our search, which itself may have excluded mentions of the concept that was expressed in ways we did not anticipate. A document-centric approach cannot tell us how RPT committees use collegiality or related concepts, if at all, during the process, nor how candidates use these guidelines in preparing their dossiers. The stated practices in these guidelines versus their actual application, or not, as well as the lived experiences of candidates and RPT committee members during the process, remain to be explored in future studies.

## Author Contributions

**Conceptualization:** Diane (DeDe) Dawson, Juan Pablo Alperin.

**Data curation:** Diane (DeDe) Dawson, Esteban Morales.

**Formal analysis:** Diane (DeDe) Dawson, Esteban Morales, Juan Pablo Alperin.

**Funding acquisition:** Erin C. McKiernan, Lesley A. Schimanski, Meredith T. Niles, Juan Pablo Alperin.

**Methodology:** Diane (DeDe) Dawson, Erin C. McKiernan, Lesley A. Schimanski, Meredith T. Niles.

**Supervision:** Juan Pablo Alperin.

**Visualization:** Juan Pablo Alperin.

**Writing – original draft:** Diane (DeDe) Dawson, Juan Pablo Alperin.

**Writing – review & editing:** Diane (DeDe) Dawson, Esteban Morales, Erin C. McKiernan, Lesley A. Schimanski, Meredith T. Niles, Juan Pablo Alperin.

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
