## [Decision Letter · Decision Letter 0]

4 Feb 2022

PONE-D-21-38592The role of collegiality in academic review, promotion, and tenurePLOS ONE

Dear Dr. Alperin,

Thank you for submitting your manuscript to PLOS ONE. After careful consideration, we feel that it has merit but does not fully meet PLOS ONE’s publication criteria as it currently stands. Therefore, we invite you to submit a revised version of the manuscript that addresses the points raised during the review process.

We look forward to receiving your revised manuscript.

Kind regards,

Mehmet Serkan Kirgiz

Academic Editor

PLOS ONE

Journal Requirements:

"Funding for this project was provided to JPA, MTN, ECM, and LAS from the Open Society Foundations (OR2018-46345). The funders had no role in study design, data collection and analysis, decision to publish, or preparation of the manuscript."

We note that you have provided funding information. However, funding information should not appear in the Funding section or other areas of your manuscript. We will only publish funding information present in the Funding Statement section of the online submission form. 

"Funding for this project was provided to JPA, MTN, ECM, and LAS from the Open Society Foundations (OR2018-46345). The funders had no role in study design, data collection and analysis, decision to publish, or preparation of the manuscript."

"MTN is a member of the board of directors of The Public Library of Science (PLOS). This role has in no way influenced the outcome or development of this work or the peer-review process, nor does it alter our adherence to PLOS ONE policies on sharing data and materials."

Additional Editor Comments :

After revising the paper to the comments written by the reviewers, the author had better to resubmit the paper to Plos One.

Reviewers' comments:

Reviewer's Responses to Questions

**Comments to the Author**

1. Is the manuscript technically sound, and do the data support the conclusions?

Reviewer #1: Yes

Reviewer #2: Yes

2. Has the statistical analysis been performed appropriately and rigorously? 

Reviewer #1: Yes

Reviewer #2: Yes

3. Have the authors made all data underlying the findings in their manuscript fully available?

Reviewer #1: Yes

Reviewer #2: Yes

4. Is the manuscript presented in an intelligible fashion and written in standard English?

Reviewer #1: No

Reviewer #2: Yes

5. Review Comments to the Author

Reviewer #1: 

1. What is the originality of this study? The authors should clearfy the originality compared with other studies.

2. Research questions, that drive the paper, should be built in the introduction from an ongoing and pertinent bibliography (up to 2021).

3. Answer your research question in the conclusions; what did we learn compared with current, significant research (up to 2021). The authors should make explicit suggestions about how their study effects.

4. The research problem is not clear, and needs to be highlighted in light of previous studies and reports.

5. Explain how this paper differs from the related ones published in the technical literature.

Reviewer #2: 

The paper gives a very good introduction to the main subject of the study (collegiality). The introduction gave account of chronological events that has necessitated the consideration of collegiality in the RPT processes.

The objectives are sound and clearly stated.

The research designed used is appropriate for the types of study. It is certain that such methodology enables the authors to carry out the study rigorously. Methods and procedures used for data collection and gathering were well explained with evidence of field documents. The sampling methods and selection were well explained, with details describing each and every stage of the process. The sample sizes used are appropriate and enough to support sound analysis. The description of the methodology made it easier for the reader to follow through. The various techniques used in the process were useful and relevant to the type of study.

Furthermore, the presentation of results followed acceptable standards and the analysis also were statistically sound. The organization of results followed the order of the research questions, which would help the reader to make easy connections among them. All the research questions were answered from the data and the analysis presented.

The conclusions are drawn appropriately from the data presented.

Lastly, the paper also acknowledged and discusses the limitations to the study and outlined measured took to reduce the impact of such limitation on the validity of the study.

Overall, the paper is well written and deserves to be accepted for publication

---

## [Editor Report · Decision Letter 1]

3 Mar 2022

The role of collegiality in academic review, promotion, and tenure

PONE-D-21-38592R1

Dear Dr. Alperin,

We’re pleased to inform you that your manuscript has been judged scientifically suitable for publication and will be formally accepted for publication once it meets all outstanding technical requirements.

Kind regards,

Mehmet Serkan Kirgiz

Academic Editor

PLOS ONE

---

## [Editor Report · Acceptance letter]

14 Mar 2022

PONE-D-21-38592R1 

The role of collegiality in academic review, promotion, and tenure 

Dear Dr. Alperin:

I'm pleased to inform you that your manuscript has been deemed suitable for publication in PLOS ONE. Congratulations! Your manuscript is now with our production department. 

Kind regards, 

on behalf of

Professor Dr. Mehmet Serkan Kirgiz 

Academic Editor

PLOS ONE